# VDAC1-Based Peptides as Potential Modulators of VDAC1 Interactions with Its Partners and as a Therapeutic for Cancer, NASH, and Diabetes

**DOI:** 10.3390/biom14091139

**Published:** 2024-09-09

**Authors:** Anna Shteinfer-Kuzmine, Manikandan Santhanam, Varda Shoshan-Barmatz

**Affiliations:** 1National Institute for Biotechnology in the Negev, Ben-Gurion University of the Negev, Beer Sheva 84105, Israel; 2Department of Life Sciences, Ben-Gurion University of the Negev, Beer Sheva 84105, Israel; santhana@post.bgu.ac.il

**Keywords:** apoptosis, cancer, mitochondria, peptide, VDAC1

## Abstract

This review presents current knowledge related to the voltage-dependent anion channel-1 (VDAC1) as a multi-functional mitochondrial protein that acts in regulating both cell life and death. The location of VDAC1 at the outer mitochondrial membrane (OMM) allows control of metabolic cross-talk between the mitochondria and the rest of the cell, and also enables its interaction with proteins that are involved in metabolic, cell death, and survival pathways. VDAC1′s interactions with over 150 proteins can mediate and regulate the integration of mitochondrial functions with cellular activities. To target these protein–protein interactions, VDAC1-derived peptides have been developed. This review focuses specifically on cell-penetrating VDAC1-based peptides that were developed and used as a “decoy” to compete with VDAC1 for its VDAC1-interacting proteins. These peptides interfere with VDAC1 interactions, for example, with metabolism-associated proteins such as hexokinase (HK), or with anti-apoptotic proteins such as Bcl-2 and Bcl-xL. These and other VDAC1-interacting proteins are highly expressed in many cancers. The VDAC1-based peptides in cells in culture selectively affect cancerous, but not non-cancerous cells, inducing cell death in a variety of cancers, regardless of the cancer origin or genetics. They inhibit cell energy production, eliminate cancer stem cells, and act very rapidly and at low micro-molar concentrations. The activity of these peptides has been validated in several mouse cancer models of glioblastoma, lung, and breast cancers. Their anti-cancer activity involves a multi-pronged attack targeting the hallmarks of cancer. They were also found to be effective in treating non-alcoholic fatty liver disease and diabetes mellitus. Thus, VDAC1-based peptides, by targeting VDAC1-interacting proteins, offer an affordable and innovative new conceptual therapeutic paradigm that can potentially overcome heterogeneity, chemoresistance, and invasive metastatic formation.

## 1. Overview

This review presents the development and activity of cell-penetrating peptides derived from the mitochondrial gatekeeper VDAC1, focusing on their role in modulating its interaction with partner proteins. VDAC1 interacts with a variety of proteins that regulate the integration of mitochondrial functions with other cellular activities. These proteins include those involved in metabolism, such as hexokinase (HK), anti-apoptotic proteins like Bcl-2 and Bcl-xL, and cytoskeletal proteins such as actin and tubulin [1,2]. The developed VDAC1-derived peptides disrupt these interactions in cancer cells, resulting in impaired energy production, altered cell metabolism, and reduced anti-apoptotic proteins [3,4,5,6,7].

The exploration of VDAC1-based peptides for disrupting protein–protein interactions (PPIs) aligns with recent advancements in targeting PPIs as a therapeutic strategy. These interactions have pivotal roles in life processes and represent a vast class of therapeutic targets both inside and outside of the cell. Several inhibitors targeting PPIs have advanced to clinical trials, along with investigations of their impact on cellular function [8,9,10]. Since PPIs involve large, flat interaction surfaces with no well-defined binding pockets as serving for substrates and ligands, small molecules face challenges in modulating these interactions effectively. Peptides appear as an important class of PPI inhibitors, as they can represent the PPI interaction site, and offer a promising alternative due to their stability, reduced toxicity, and lower immunogenicity compared to traditional chemotherapeutic agents used in cancer treatment.

In this review, we highlight the development of VDAC1-derived cell-penetrating peptides and their activity in cell culture and in disease mouse models. These peptides, by targeting VDAC1-interacting proteins, offer an innovative new conceptual therapeutic paradigm that can overcome tumor heterogeneity, chemo resistance, and invasive metastatic formation.

## 2. VDAC1 as a Hub Protein—Modulation of Its Interactome to Target Apoptosis, Metabolism, and Inflammation

VDAC1 is a mitochondrial gatekeeper that has multiple protein functions, and it is a convergence point for a variety of cell survival and death signals. VDAC1, located at the OMM, mediates cross-talk between the mitochondria and other parts of the cell by transporting metabolites, fatty acids, anions, cations, ATP, Ca^2+^, and other molecules up to 5 kDa [1,2]. Most cancer cells rely on glycolysis as the main pathway for generating energy (Warburg effect) and as a source of products for generating proteins, nucleotides, and lipids [11]. Such metabolic re-programming in cancer cells also includes a marked overexpression of VDAC1 [12,13,14]. It has also been implicated in mitochondrial lipid import [15,16,17].

Substantial evidence points to VDAC1 as being a key player in mitochondria-mediated apoptosis, regulating the release of apoptogenic proteins such as cytochrome c (Cyto c) from the mitochondria, and interacting with anti-apoptotic proteins [1,7,18,19,20,21]. Through its oligomerization, it forms a large channel that allows the release of pro-apoptotic proteins [1,2,19,22,23] and mitochondrial DNA [24] to the cytosol, which results in apoptosis and inflammation. VDAC1 also regulates apoptosis by serving as the anchoring site for several anti-apoptotic proteins, including HK, Bcl-2, and Bcl-xL [18,20,21,25,26,27,28,29]. Most tumor cells have also developed apoptosis escape mechanisms that involve upregulation of HK and the Bcl-2 family. Mitochondrial-bound HK and Bcl-2 are overexpressed in many cancer cells, including breast, lung, pancreas, esophagus, renal, and liver cancers, while Bcl-2 is overexpressed in colon, breast, prostate, lymphoma, glioma, and leukemia cells, and their overexpression in tumors is coupled with resistance to chemotherapy-induced apoptosis [30,31].

VDAC1 is crucial for many cellular processes, including metabolism, Ca^2+^ homeostasis, apoptosis, and inflammation. VDAC1 is a hub protein that interacts with over 150 proteins that regulate the integration of mitochondrial functions with other cellular activities associated with cell survival and cellular death pathways [1,2]. VDAC1 interactome includes proteins involved in metabolism, apoptosis, signal transduction, anti-RNA-associated proteins, and more [1,2] (Figure 1). VDAC1 serves as an anchor protein for proteins located in the OMM, inner mitochondrial membrane (IMM), intermembrane space (IMS), cytosol, ER, plasma membrane, or nucleus [1,2] (Figure 1). Thus, VDAC1 appears to be a convergence point for a variety of cell survival and death signals, mediated via an association with ligands and proteins. The variety of proteins interacting with VDAC1 and the bidirectional effects of VDAC1 on its interacting proteins and vice versa are presented in the following subsections.

### 2.1. VDAC1 Interaction with Metabolism-Related Proteins

VDAC1 displays binding sites for a large number of metabolism-related proteins such as glycerol kinase (GK), HK, C-Raf kinase, adenine nucleotide translocase (ANT) [32], and the glycolytic enzyme glyceraldehyde 3-phosphate dehydrogenase (GAPDH) [33]. Mitochondrial creatine kinase (MtCK), in its octameric state, interacts with VDAC1 [34] and causes decreased VDAC1 affinity for HK and Bax [35]. HK binding to VDAC1 [21,36,37,38,39] allows mitochondrial generated ATP to directly couple with glucose phosphorylation. Thus, the formation of a VDAC1–HK complex coordinates glycolytic flux with that of the TCA cycle and ATP synthase [40,41].

The OMM protein carnitine palmitoyltransferase 1a (CPT1a), which catalyzes the primary step of fatty acid oxidation, also interacts with VDAC1 [42,43]. VDAC1 was found to be part of a complex mediating the transport of fatty acids through the OMM in rat liver mitochondria [42]. VDAC1 acts as an anchor, linking the long-chain acyl-CoA synthetases (ACSLs) at the OMM to CPT1a, which faces the IMS. According to the proposed model, upon activation by ACSLs, VDAC1 transfers acyl-CoAs across the OMM to the IMS, where they are converted into acylcarnitine by CPT1a. VDAC1, CPT1a, and ACSL1 were co-purified along with CPT2, localized in the IMM [43,44]. These interactions were further validated using an in situ proximity ligation assay (PLA) [45], demonstrating interactions between VDAC1-CPT1a, CPT1a-ACSL1, and VDAC1-ACSL1 [43].

At the OMM, VDAC1 interacts with and modulates the function of the translocator protein TSPO [46,47], which is involved in the transport of cholesterol into the mitochondria [48]. TSPO-VDAC1 interaction affects TSPO’s ligand-binding characteristics [49] and contributes to mitochondrial quality control. The interaction between TSPO and VDAC1 is considered to play a role in the activation of the mitochondrial apoptosis pathway, with the grouping of TSPO molecules around VDAC1 [49,50] and increased ROS generation in the proximity of the VDAC1, leading to apoptosis induction [47,50].

### 2.2. VDAC1 Interaction with Apoptosis-Related Proteins

Cancer cells are able to suppress pro-apoptotic pathways and/or activate anti-apoptotic mechanisms [11,51] that are associated with drug resistance [52].

The Bcl-2 family comprises pro-apoptotic (e.g., Bid, Bax, Bim, and Bak) and anti-apoptotic (e.g., Bcl-2 and Bcl-xL) members that stimulate or inhibit apoptosis, respectively [18,20,26,53,54,55,56,57]. Bcl-2 and Bcl-xL were found to interact with bilayer-reconstituted VDAC1, leading to a decrease in the channel conductance of native VDAC1, but not its mutated form [20]. Additionally, these proteins provided protection against apoptosis in cells expressing native VDAC1, but not in those with the mutated version [18,20,27,58,59].

Mcl-1 was shown to directly interact with VDAC1 to increase mitochondrial Ca^2+^ uptake and ROS generation [60]. In addition, the pro-apoptotic protein BNIP3 was shown to interact with VDAC1 to induce mitochondrial release of endonuclease G [61]. The VDAC1 domains that are involved in the interaction with Bcl-2 and Bcl-xL to confer anti-apoptotic activity were identified by site-directed mutagenesis [18,20].

The HK–VDAC1 interaction also prevents release of pro-apoptotic factors, such as Cyto c, and its detachment subsequently, induces apoptosis [21,36,37,38,39]. This interaction was prevented by VDAC1-based peptides, decreasing ATP production, glycolysis, and apoptosis through HK detachment [21,28,29,35,36,39,40].

Since the anti-apoptotic proteins HK-I, HK-II, Bcl-2, and Bcl-xL have been found to be expressed at high levels in many types of cancer [40,62,63,64,65,66,67], interfering with their interaction with VDAC1 is an appropriate target for inducing apoptosis. Thus, VDAC1-based peptides targeting the anti-apoptotic proteins provide an opportunity to develop new anti-cancer therapies that allow the chemo resistance of cancer cells to be overcome. 

### 2.3. VDAC1 Interaction with Cytoskeletal Proteins

VDAC1 interacts with several cytoskeletal proteins, such as gelsolin, in an interaction that results in inhibited VDAC1 channel activity and Cyto c release from liposomes in a Ca^2+^-dependent manner [68,69].

Tubulin associates with VDAC1 [70] and induces VDAC1 closure [71]. Dimeric αβ-tubulin is proposed to play a regulatory role in the cellular bioenergetics of skeletal and heart muscles and brain synaptosomes through interaction with VDAC1, with monomers of αβ-tubulin decreasing the passage of ATP through the channel [71,72,73,74]. It is also proposed that tubulin sustains the Warburg effect [75], and that tubulin, VDAC1, and MtCK form a super-complex that is structurally and functionally coupled to the ATP synthasome [76]. It has been proposed that the dynamic changes in ΔΨm brought about by free tubulin in tumor cells are related to αβ-tubulin heterodimers that modulate VDAC1 conductance [77].

G-actin directly and selectively binds to VDAC in yeast [78], and reduces the conductance of the Neurospora crassa VDAC channel [79]. It has been shown that a dynamic actin cytoskeleton is required for preventing constant VDAC-dependent MAPK signaling at the mitochondria and for maintaining proper lipid homeostasis [17]. 

Microtubule-associated protein 2 (MAP2) was shown to bind VDAC [80]. Interaction of VDAC1 with Tctex-1/dynein light chain 1 (DYNLT) has also been demonstrated [81].

The D-Δ(1–18)N-Ter-Antp VDAC1-N-terminus-derived peptide was found to increase actin and tubulin expression and alter their organization, and induce reorganization of actin and tubulin filaments [3]. It reduced the transverse actin stress-fiber arcs, ventral and dorsal stress fibers, and fine actin filaments in the cytosol, and caused the development of strong filopodia.

### 2.4. VDAC1 Interactions with Signaling Proteins

VDAC1 interacts with several cell signaling proteins. Among these are the L-type Ca^2+^ channel [82] and inositol 1,4,5-trisphosphate receptors (IP3Rs). VDAC selectively co-immuno-precipitates with IP3R [83]. VDAC1, IP3R, and glucose-regulated protein 75 (GRP 75) interact at the endoplasmic reticulum (ER)–mitochondria contact site, enabling Ca^2+^ transfer into the mitochondria [84]. A VDAC1-based peptide was found to disrupt Ca^2+^ homeostasis, increasing intracellular [Ca^2+^]i levels [5].

Superoxide dismutase (SOD1) is a predominantly cytosolic protein, with mutant SOD1 being present mostly in fractions enriched for mitochondria [85,86,87]. Mutant SOD1, associated with amyotrophic lateral sclerosis (ALS), bound to bilayer-reconstituted VDAC1 and inhibited its channel conductance [88]. Studies have demonstrated that the interaction between mutant SOD1 and Bcl2 leads to mitochondrial dysfunction. However, a small peptide derived from Bcl2 can restore mitochondrial function and cell viability by modifying Bcl2’s interaction with VDAC1 [89].

The VDAC1 N-terminus-derived peptide interacted with SOD1 and modulated mitochondria-mediated apoptosis while protecting against the death of motor neuron-like NSC-34 cells that expressed mutant SOD1 [6].

Endothelial NO synthase (eNOS) was also found to bind VDAC1. Such interaction amplified eNOS activity in an intracellular Ca^2+^-mediated manner [90]. This interaction is important for regulating eNOS activity and modulating VDAC1 [90].

Mitochondrial anti-viral signaling protein MAVS localized in the OMM [91] was demonstrated to mediate its pro-apoptotic activity via VDAC1 and to modulate VDAC1 protein stability via the ubiquitin–proteasome pathway [92]. Recently [93], we demonstrated that MAVS directly interacts with VDAC1 with high affinity (950 nM), decreasing its conductance via interaction with the cytosolic face of VDAC1. The direct association of MAVS with VDAC1 in the cell was shown using a PLA, with this interaction demonstrating regulation of mitochondria-dependent MAVS activity [93].

VDAC1-interacting protein complexes mediate and/or regulate metabolic, apoptotic, and other processes that may be impaired under disease conditions.

VDAC1-based peptides that can interfere with these interactions and lead to impaired cell metabolism, apoptosis, and inflammation [3,4,5,6,7,18,20,21,43] are presented here.

## 3. Development of Cell-Penetrating, Stable, and Effective VDAC1-Based Peptides

Given that VDAC1 can interact with so many proteins, we developed VDAC1-based peptides as a “decoy” to compete with VDAC1 for these proteins at the HK–, Bcl-2–, and Bcl-xL–VDAC1 interaction sites and, consequently, were able to interrupt their anti-apoptotic pro-survival activity which is associated with cancer [4,5,7,18,20,21,28]. Via point mutations, we identified VDAC1 domains and amino acid residues that are important for interactions with HK, Bcl-2 and Bcl-xL and SOD, and cell penetrating VDAC1-based peptides targeting these interactions were designed and tested [3,4,5,6,7,18,20,21,28,43].

The structure of mammalian VDAC1 was determined at atomic resolution, revealing that it is composed of 19 transmembrane β-strands that are connected by flexible loops to form a β-barrel, with strands β1 and β19 in a parallel conformation along with a 26-residue-long N-terminal domain (NTD) that lies inside the pore (Figure 2A). We were able to identify two VDAC1-based sequences: the VDAC1-N-terminus and a loop within the VDAC1 β-strands that was found to mediate the interaction of VDAC1 with its partners (Figure 2B).

### 3.1. The VDAC1 N-Terminal-Derived Peptide

Several different functional roles for the N-terminal domain have been proposed, including acting as a voltage sensor that regulates channel gating and conductance of ions and metabolites that pass through the pore [54] and possession of an ATP-binding site.

The N-terminal domain is required for the release of Cyto c and subsequent activation of apoptotic cell death as evidence from the finding that no cell death was induced in cells expressing Δ(1–26)VDAC1. In addition, Δ(1–26)VDAC1 showed no voltage-dependent conductance and exhibited high conductance at all membrane potentials tested [28]. Importantly, N-terminal domain mobility is required for VDAC1 dimer formation [54], as well as for the interaction of apoptosis-regulating proteins of the Bcl-2 family (i.e., Bax, Bcl-2, and Bcl-xL) [18,20,26,28,54], HK-I and HK-II [21,28] and SOD1 [6], and amyloid beta (Aβ) [94]. It modulated mitochondria-mediated apoptosis while protecting against the death of motor neuron-like NSC-34 cells that express mutant SOD1 [6].

Due to the importance of the VDAC1-N-terminal’s functions, several CPPs derived from it were designed (Figure 3), with the aim of identifying the shortest peptide with improved cellular stability and activity.

The N-terminal domain (NTD) (26 amino acids in length) was produced at different lengths, amino acids 1–4, 1–10, 1–14, 1–18, and from the C-terminus 21–26, with the aim of identifying the shortest peptide with cell death activity [3]. To allow peptide penetration into the cell, it was fused to the 16-residue-long sequence from the Drosophila antennapedia homeodomain, Antp, also known as penetrating [95], as a cell-penetrating peptide (CPP). To enhance N-Ter-Antp peptide stability, L-amino acids were replaced with an unnatural D-amino acid configuration that increases peptide cellular stability and resistance to proteases. The NTD version that was shortened from the N-terminus by 18 amino acids, comprising amino acids 19–26, in D confirmation and fused to Antp yielded a D-Δ(1–18)N-Ter-Antp peptide that resulted in a cell death-inducing peptide [3,5,7].

On the other hand, deletion of the sequence containing the GXXXG motif, known to participate in protein–protein interactions, yielded an N-Ter Δ(21–26)-Antp peptide that exhibited highly reduced cell death activity [3,5]. The GXXXG motif has been implicated in α-helical structures of the dimer formation of membranal proteins [96].

The VDAC1-N-terminal lies inside the pore and can be translocated out of it [54]. The flexibility required for its translocation out of the channel pore is believed to involve a flexible glycine-enriched sequence (21GYGFG25) that connects the N-terminal domain to β-strand 1 of the barrel [54].

In addition to cell death, the peptide induced alterations in the expression of proteins associated with cell metabolism, signaling, and division such as enhancing the expression of nuclear factor kappa B and decreasing the expression of nuclear factor of kappa light polypeptide gene enhancer in B-cell inhibitor alpha [3]. The peptide also induced multiple effects, including apoptosis, autophagy, senescence (Figure 4), adhesion, and altering cell division. Moreover, it induced the re-fusion of divided daughter cells into a single cell, and also promoted the re-organization of actin and tubulin filaments (Figure 4 and Figure 5).

### 3.2. The VDAC1-Derived Peptide R-Tf-D-LP4

Because the loop between β-strands 14 and 15 was found to be involved in the association of VDAC1 with HK, Bcl-2, and Bcl-xL [18,20,21,28,36,37], it was selected for development of a VDAC1-based peptide to compete with VDAC1 binding with these and other proteins.

A loop-shaped peptide composed of a VDAC1-derived sequence representing the loop between β-strands 14 and 15 (defined as LP4) was stabilized as a loop by the tryptophan zipper motif (Figure 3B). To allow its penetration into the cell, several CPPs were used. Either the Antp [95], HIV-1 TAT [97], or the transferrin receptor (TfR) internalization sequence HAIYPRH (Tf) [98] were conjugated to the VDAC1-derived peptide to yield a cell-penetrating peptide.

In order to optimize the length of the peptides and to enhance their plasma stability and specific targeting of cancer cells, over 40 versions of these peptides were designed and evaluated [4,5,7].

Of the various cell-penetrating LP4-based peptides, the TAT-conjugated peptide was less effective in inducing cancer cell death. The Tf-conjugated D-LP4 peptide, Tf-D-LP4, was highly effective in inducing cell death more than the Antp-conjugated peptides. The greater effectiveness of the Tf-D-LP4 peptide led us to construct a retro-inverso peptide (R-Tf-D-LP4) [5,7] (Figure 3D).

The retro-inverso isomer of an LP4-peptide is composed entirely of D-amino acid residues with the direction of the peptide bonds reversed to maintain a side-chain topology similar to its parent L-molecule, but with inverted amide peptide bonds [99]. Retro-inverso peptides retain the key structural features of the peptide backbone, and this isomeric change is considered functionally neutral due to similarity at the primary structural level [100]. These peptides are extensively utilized in developing proteolytically stable D-isomers that replicate the biological activities of their parent L-peptides. They generally exhibit enhanced stability, as evidenced in various peptides such as enkephalin, glutathione, Substance P, gastrin, and atrial natriuretic peptides [101].

R-Tf-D-LP4 exhibited superior cell death-inducing activity compared to the Antp-LP4 peptides. This enhanced efficacy may stem from its improved stability and enhanced cell penetration, given the high expression of TfR in cancer cells [98]. Additionally, R-Tf-D-LP4 was observed to inhibit HK activity effectively and showed greater solubility and stability [5,7].

The Tf-D-LP4 peptide has been shown to induce apoptosis in several cancer types and in mouse models of glioblastoma [7], and lung and breast cancers [5]. It was found to simultaneously attack several hallmarks of cancer: impaired energy and metabolic homeostasis, inhibited tumor growth, cell proliferation and invasion, and induced apoptosis. The Tf-D-LP4 caused overexpression of apoptotic proteins, and it eliminated cancer stem cells from which cancer can re-develop [5,7] (Figure 5). Furthermore, the peptide Tf-D-LP4 was found to be effective in treating non-alcoholic fatty liver disease [15] and type 2 diabetes [102].

In summary, the outcomes of our extensive peptide development process (Figure 3) led to the identification of three optimized cell-penetrating VDAC1-based peptides: D-∆(1–18)-N-Ter-Antp, Tf-D-LP4, and R-Tf-D-LP4. These peptides were evaluated for their ability to induce apoptosis in lymphocytes derived from patients with chronic lymphocytic leukemia (CLL) and in other cell types, and evaluated in vivo in glioblastoma, breast, and lung cancer mouse models, and found to induce cell death and to inhibit tumor growth [3,4,5,7].

## 4. Proposed Mode of Action of VDAC1-Based Peptides

The VDAC1-based peptides demonstrated a multi-mode of action, including energy and metabolism impairment, interference with the action of anti-apoptotic proteins, and triggering of senescence and cell death (Figure 5). These multi-effects of the peptide possibly are mediated via interference with VDAC1-interacting proteins, which thereby influence multiple mitochondrial/VDAC1 pathways that are associated with cell functions. The mode of action of the VDAC1-based peptides (Figure 5) involves several pathways: (i) impairment of cell metabolism and energy homeostasis [3,5,21,28]; (ii) interaction with the anti-apoptotic activities of HK, Bcl-2, and Bcl-xL, which promote apoptosis [28,36]; (iii) elimination of CSCs [4,5,7,28,36]; and (iv) induction of senescence [3] (Figure 4).

### 4.1. Impairment of Cell Metabolism and Energy Homeostasis

A hallmark of many malignant tumors is altered energy metabolism, particularly using glycolysis as an energy source [11,103]. Glycolytic cancer cells often exhibit elevated levels of mitochondrial-bound HK, which supports aerobic glycolysis [40] and confers resistance to apoptosis [21,29,35,36,39,104]. Peptides derived from VDAC1 were found to disrupt cell metabolism and energy homeostasis. By binding to VDAC1, HK gains direct access to mitochondrial ATP and enhances glycolytic activity. As previously discussed, VDAC1 is frequently overexpressed in various cancer types [12], providing docking sites for the overexpressed HK, and thereby facilitating glycolysis. VDAC1-based peptides interact with HK and displace it from its binding site on the VDAC1, resulting in reduced glycolysis, decreased mitochondrial membrane potential (ΔΨ), and lower cellular ATP levels [21,28]. Consequently, detachment of HK from the mitochondria disrupts the overall cellular bioenergetic balance that is crucial for sustaining the high energy demands of cancer cells.

### 4.2. Inhibiting the Anti-Apoptotic Activities of HK, Bcl-2, and Bcl-xL and Promoting Apoptosis

Cancer cells employ various strategies to avoid apoptosis, including the overexpression of anti-apoptotic proteins such as members of the Bcl-2 family and HK, and VDAC1 binds HK, Bcl-2, and Bcl-xL [18,20,21,25,26,27,28,29,54]. VDAC1-based peptides were found to interact directly with purified anti-apoptotic proteins and, upon entering cells, to disrupt the interaction between the anti-apoptotic proteins and VDAC1, counteracting their anti-apoptotic effects, and thereby promoting apoptosis [4,18,20,21,28,36].

It was demonstrated that VDAC1 oligomerization is a dynamic process, and it represents a general mechanism common to numerous apoptotic stimuli: acting through different initiating cascades [19,22,23,105,106,107,108,109,110]. In addition, VDAC1-based peptides induce VDAC1 oligomerization, forming large pores that allow the release of mitochondrial pro-apoptotic proteins, thereby promoting apoptosis (Figure 5).

### 4.3. Elimination of Cancer Stem Cells (CSCs)

The CSC hypothesis postulates that a subpopulation of malignant cells constantly supplies a tumor with cancerous cells. CSCs, as embryonic and somatic stem cells, have self-renewal and multi-potent differentiation abilities [111,112]. Treatment of tumors with the Tf-D-LP4 peptide effectively eliminated these cells in glioblastoma, lung, and breast cancers, as evidenced by significant reductions in the expression levels of cancer type-specific stem cell markers [5]. Because CSCs are relatively quiescent and resistant to chemotherapy and radiation [113,114,115], they are a potentially promising target for therapeutic intervention. Thus, VDAC1-derived peptide anti-CSCs can also account for their high efficacy against tumor cells.

### 4.4. Inducing Senescence

Cells treated with VDAC1-based peptides displayed cellular senescence [3] (Figure 4A), exhibited by somatic cells losing their capacity to proliferate after a limited number of mitotic divisions and entering cell cycle arrest [116]. This senescence is induced by various stressors [117]. 

The multiple mode of peptide activities may explain their high potency and specificity toward tumor cells.

## 5. VDAC1-Based Peptides Targeting Diseases

VDAC1 functions as the mitochondria gatekeeper and regulates ATP production, Ca^2+^ homeostasis, and apoptosis execution, which are indispensable for proper mitochondrial function and, consequently, for normal cell physiology. Therefore, it is not surprising that VDAC1 is associated with a range of diseases. The effects of VDAC1-based peptides are detailed below.

### 5.1. Targeting Cancer with VDAC1-Derived Peptides

Cancer is a complex disease in which cells acquire a common set of properties, including unlimited proliferation, metabolic reprogramming, and resistance to anti-proliferative and apoptotic cues [11,51]. As the mitochondria gatekeeper, VDAC1 is associated with these activities and interacts with a large number of proteins that are associated with these cell functions [1,2]. Targeting these interactions with VDAC1-based peptides resulted in impaired energy production and activation of apoptosis in cancerous tissue (Figure 5).

CLL is characterized by a clonal accumulation of mature neoplastic B cells and is resistant to apoptosis. CLL cells express high levels of anti-apoptotic proteins [4]. In an ex vivo study, cell-penetrating VDAC1-based peptides were found to induce apoptotic cell death in B-cells in peripheral blood mononuclear cells (PBMCs) obtained from CLL patients, yet spared those obtained from healthy donors [4], pointing to the potential of VDAC1-based peptides as an innovative and effective anti-CLL therapy.

In subcutaneous xenograft mouse models of glioblastoma (GBM), lung cancer, and triple negative breast cancer (Figure 6), as established using U-87MG, A549, and MDA-MB-231 cells, respectively, intratumoral treatment with R-Tf-D-LP4, Tf-D-LP4, or D-ΔN-Ter-Antp peptides highly inhibited tumor growth (~70–90%) [5,7]. The peptides inhibited cell proliferation, reduced energy production, and induced apoptosis [5,7]. Moreover, they reduced the expression levels of Glut 1, HK-I, HK-II, GAPDH, and LDH and of proteins active in Kreb’s cycle OXPHOS. This suggests that, in the residual peptide-treated tumor, the re-programmed metabolism of the cancer cell was reversed.

In the three types of cancer studied, glioblastoma, lung, and breast cancers, the peptides induced massive apoptotic cell death and induced the overexpression of key proteins in apoptosis, including caspases 3, caspase 8, p53, Cyto c, and SMAC/Diablo, while decreasing the expression of anti-apoptotic proteins, such as Bcl-2 [5,7]. Furthermore, the peptides eliminated CSCs in tumors derived from either glioblastoma, lung, or breast cancers [5,7], which are relatively quiescent and resistant to chemo- and radiotherapies [115].

Moreover, the R-Tf-D-LP4 was given intravenously (i.v.) in an intracranial–orthotopic xenograft GBM mouse model [7], comparing the effects of the peptide when administered. Tf-D-LP4 was found to highly reduce tumor volume, and it simultaneously attacked several cancer hallmarks, causing impairment of energy and metabolic homeostasis, induction of apoptosis, and overexpression of apoptotic proteins [7]. The effect of the i.v.-administered peptide on the tumor in the brain suggests that it crosses the blood–brain barrier (BBB). This is expected as R-Tf-D-LP4 contains the cell-penetrating peptide Tf, recognized by TfR, that has been shown to be enriched in the BBB [118,119].

To conclude, the multifactorial effects of the peptide interfering with protein–protein interactions involving VDAC1, as summarized in Figure 5, clearly reflect the importance of VDAC1 interactions with many proteins [1,2] that are associated with metabolism, apoptosis, and other functions important for cancer cell growth and tumor development.

### 5.2. Targeting NASH with the VDAC1-Derived Peptide R-Tf-D-LP4

Non-alcoholic fatty liver disease (NAFLD) is a leading cause of chronic liver disease in Western countries (20–30%) and is the growing cause of hepatocellular carcinoma (HCC) [120]. Prevalence is further increased in patients with type 2 diabetes (T2D) [121], and is commonly associated with obesity and cardiovascular disease [122]. NAFLD is characterized by an excessive abnormal accumulation of fatty acids and triglycerides within the hepatocytes of non-alcohol users. The progressive start of the disease is non-alcoholic steatohepatitis (NASH) [123], which represents a spectrum of disorders that include abnormal lipid metabolism, oxidative stress, lipid toxicity, mitochondrial dysfunction, inflammation, and ER stress [124], with a severe form that leads to liver fibrosis–cirrhosis and HCC [122].

Treatment of mice subjected to a high-fat diet (HFD-32) with R-Tf-D-LP4 was found to reverse the liver steatosis and NASH pathology to a normal-like state [43]. The peptide arrested steatosis and NASH progression, eliminated hepatocyte ballooning degeneration, inflammation, and liver fibrosis, and restored liver pathology-associated enzymes and glucose levels with steatosis, NASH, and HCC [43]. Peptide treatment increased the expression of enzymes and factors associated with fatty acid transport to the mitochondria, and enhanced β-oxidation and thermogenic processes, yet decreased the expression of enzymes and regulators of fatty acid synthesis. As the peptide induced no cell death, it affected carbohydrate and lipid metabolism and, consequently, offers a promising therapeutic approach for steatosis and NASH [43].

### 5.3. Targeting Diabetes Mellitus with the VDAC1-Derived Peptide R-Tf-D-LP4

Diabetes mellitus (DM) refers to chronic metabolic disorders characterized by hyperglycemia [125] as a consequence of insulin dysfunction, either due to defects in insulin secretion, insulin action, or both [126]. DM is a severe metabolic disease, and type 2 diabetes (T2D) has escalated globally in the wake of the obesity epidemic. Uncontrolled diabetes can result in cardiovascular and chronic kidney diseases, stroke, peripheral neuropathy, blindness, and potentially death [127].

In response to insulin resistance, β-cells increase insulin production, which gradually leads to their destruction, as well as to hyperglycemia and eventual insulin dependence [128].

Increased cellular lipid content has been identified as a contributing factor to insulin resistance [129], which is a common characteristic in various metabolic disorders such as obesity, hypertension, and non-alcoholic NAFLD [130,131]. Moreover, in both NAFLD and diabetes, mitochondrial dysfunction has also been implicated, contributing to insulin resistance [132,133,134].

In our NAFLD studies (Section 5.2), we utilized a mouse model induced by streptozotocin (STZ) and a high-fat diet (STZ/HFD-32) to simulate T2D and NAFLD phenotypes [43]. The R-Tf-D-LP4 peptide attenuated several parameters of an NAFLD mouse model on a diabetic background [43]. In addition, the peptide decreased elevated glucose levels in ob/ob mice.

In STZ/HFD-32-fed mice, the R-Tf-D-LP4 peptide reduced the elevated blood glucose levels to near normal, increased the insulin content in the islets, and enhanced both the number and average size of the islets compared to untreated controls [102]. Treatment with the R-Tf-D-LP4 peptide also led to increased expression of the proliferation marker KI-67 in the islets, along with enhanced expression of β-cell maturation and differentiation transcription factor PDX1. PDX1 is essential for regulating insulin gene expression and is also key to the development, function, and proliferation of pancreatic islets. This underscores the potential benefits of using the VDAC1-based peptide as a treatment for diabetes [102].

### 5.4. VDAC1-Based Peptide Targeting Levels of Testosterone

Mitochondrial cholesterol import is controlled by interactions between VDAC1, the steroidogenic acute regulator protein STAR [135], and the translocator protein TSPO [48,136,137], forming a transduceosome, which also contains the 14-3-3ϵ protein adaptor that interacts with VDAC1 [138].

The transduceosome determines the availability of cholesterol for steroidogenesis. A cell-penetrating VDAC1-based peptide composed of 25 amino acids (159–172) was found to block the interaction of 14-3-3ϵ with VDAC1 and an in vivo study showed it to significantly increase levels of testosterone [138]. More recently [139], RVTQ was identified as a minimum sequence of this peptide and was developed for oral administration. It was shown to increase circulating testosterone levels in male rats.

In essence, VDAC1 plays crucial roles in ATP production and metabolism. Moreover, Ca^2+^ homeostasis and apoptosis execution are indispensable for proper mitochondrial function and cellular vitality. These functions hinge on VDAC1’s interactions with numerous proteins that regulate cell survival and death pathways. Modulating VDAC1′s interactions with its partners, as well as proteins associated with metabolism, apoptosis, and more, using VDAC1-based peptides represents a promising avenue for developing next-generation treatments that could significantly impact the management of diverse diseases.

## 6. Conclusions

Because most of the proteins perform their cellular function through interactions with each other and other molecules, such as DNA and RNA, their protein–protein interactions (PPIs) encompass the formation of stable or transient protein complexes that are essential for normal cell functioning

Abnormal PPIs have been linked to a range of diseases, including cancer, infections, and neurodegenerative disorders [8,9]. Consequently, targeting these interactions represents a promising approach for disease treatment and is a crucial strategy in the development of new drugs. 

Due to their small size, small molecules do not effectively interact with large surfaces, such as PPIs. Typically, PPIs cover a contact area of 1500–3000 Å^2^, whereas small molecules only cover 300–1000 Å^2^ of the protein surface [140]. In contrast, peptide drugs, with their larger size and more flexible backbone, enable them to act as potent inhibitors of PPIs [10]. Moreover, peptides derived from the interaction sites between the interacting proteins represent a good strategy for targeting PPIs.

Recent advancements in molecular biology, peptide chemistry, and peptide delivery technologies have driven significant progress in peptide drug discovery, production, and therapeutic applications. Peptides are of particular interest as therapeutic drugs because they are naturally produced and have fewer side effects. However, as natural amino acid-based therapeutics, therapeutic peptides have two intrinsic drawbacks: membrane impermeability and a short half-life and fast elimination in vivo. These intrinsic advantages and disadvantages of peptides present both challenges in peptide drug development and also opportunities and directions for peptide drug design and optimization. Peptide modifications allow peptides to achieve better plasma stability, and CPPs allow cell penetration and targeting to the cytosol, mitochondria, or the nucleus [95,97,98,141,142].

Over 80 therapeutic peptides are now available on the global market, with hundreds more in pre-clinical and clinical development. These peptide drugs are being applied in a variety of diseases, including diabetes mellitus, cardiovascular disorders, gastrointestinal and infectious diseases, and cancer, and are in vaccine development [143,144].

Many PPIs have now been targeted, and several inhibitors have reached clinical trials [9]. Peptides derived from the interaction sites between the interacting proteins represent a good strategy for targeting PPIs [145].

VDAC1 plays multiple roles as a key convergence point for various cell survival and death signals, requiring interactions with numerous cellular proteins. Its interactome includes over 150 proteins, located in the OMM, IMM, IMS, cytosol, ER, plasma membrane, and nucleus, and these interactions are crucial for its regulation of cell life and death. Targeting these interactions presents a promising strategy for modulating PPIs. As detailed in this review, several groups have developed peptides derived from VDAC1’s interaction sites with its partner proteins, demonstrating their potential in targeting cancer and other diseases.

## Figures and Tables

**Figure 1 biomolecules-14-01139-f001:**
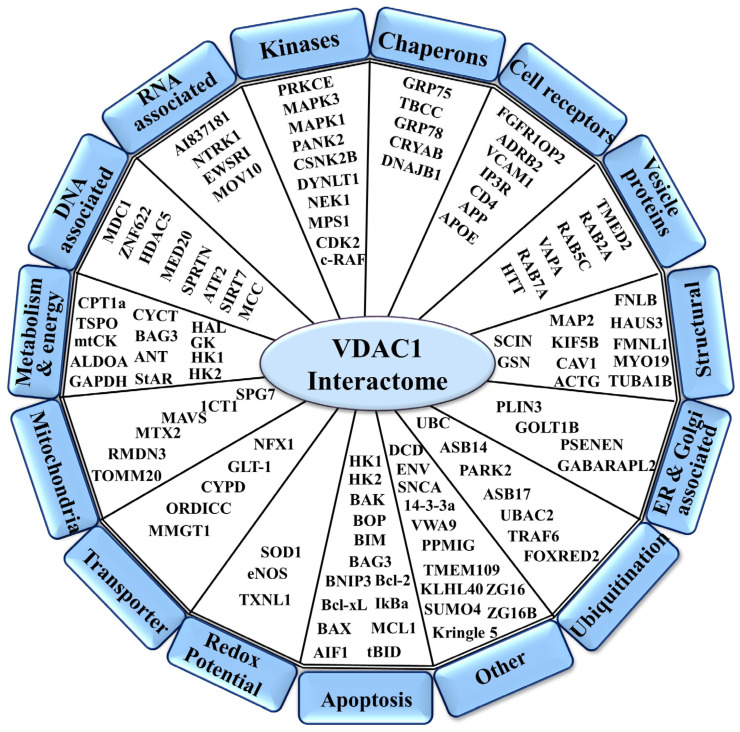
Schematic representation of VDAC1 as a hub protein with its associated interacting proteins, sub-grouped into those associated with metabolism, energy, apoptosis, anti-oxidation, cell reception, or signal transduction, and their localization to the mitochondria, ER, nucleus, and cell membrane is indicated. The full name of each protein can be found on the Abbreviations list, and their function is described in [1,2]. Proteins interacting with VDAC1 were identified using various methods, as outlined in Table 1 of Ref [1]. These methods include affinity capture of the complex and identification of the interacting protein either by mass spectrometry (MS) or immunoblotting using antibodies specific to VDAC1 and its interacting partners; co-fractionation of VDAC1 interacting proteins that were co-purified; Förster resonance energy transfer (FRET) between pairs of fluorophore-labeled molecules such as VDAC1 and any partner; microscale thermophoresis (MST), measuring the direct interaction between VDAC1 and a selected purified protein; VDAC1 reconstituted into a planar lipid bilayer, and interaction of the selected protein with VDAC1 reflected in changes in channel conductance; surface plasmon resonance with VDAC1 or the interacting protein covalently attached to a sensor chip that then was exposed to the tested protein via a microfluidic system to monitor binding interactions; and two-hybrid, with protein interactions detected through activation of a reporter gene.

**Figure 2 biomolecules-14-01139-f002:**
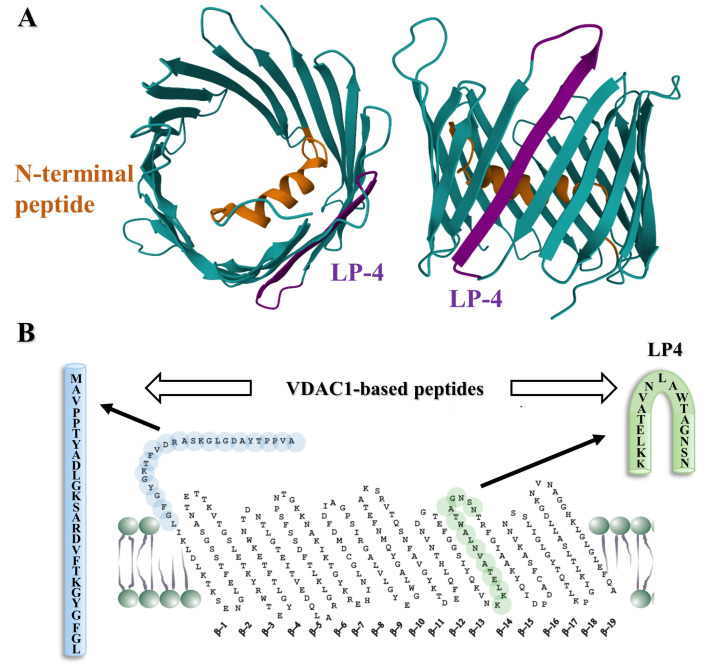
VDAC1 structure and the selected peptides for development. (**A**) VDAC1 monomer, top view and side view of the crystal structure of VDAC1 (PDB code: 3EMN). The β-barrel is formed by 19 β-strands and the N-terminal domain (orange) is folded into the pore interior. The β-strands representing the LP4 (purple) are indicated. (**B**) VDAC1-derived sequences N-terminus and loop-shaped LP4 appear in blue and green, respectively, as indicated in the VDAC1 transmembrane topology with the N-terminal domain and 19 trans-bilayer β-strands.

**Figure 3 biomolecules-14-01139-f003:**
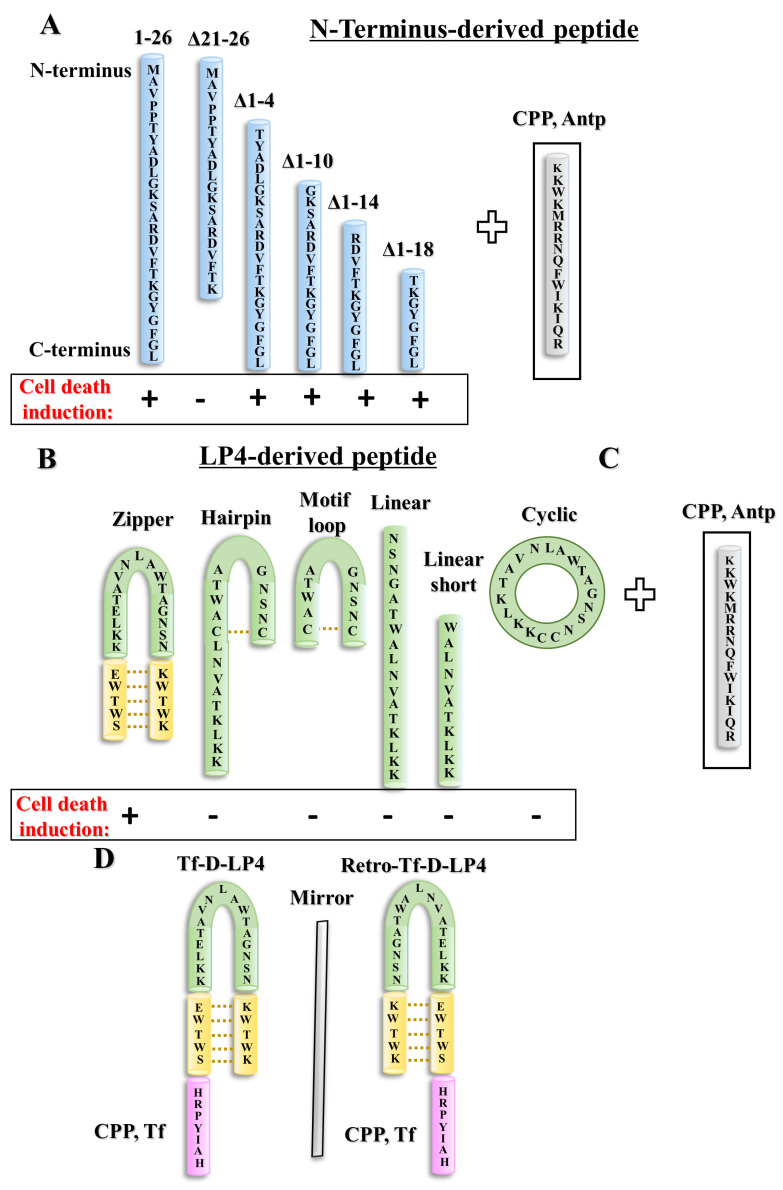
VDAC1-based peptide deployment and optimization. (**A**) The N-terminus peptide (1–26 aa) and the peptides derived from it following deletion of part of the sequence resulted in Δ(21–26), Δ(1–4), Δ(1–10), Δ(1–14), and Δ(1–18) N-terminus peptides, which are shown. Following the fusion of the CPPs: Antp (gray), the activity of various CPPs in the induction of cancer cell death is indicated at the bottom of each peptide as + (active) and − (non-active) [3]. (**B**) The structure of the LP4 peptide is composed of a zipper-loop-shaped (Trp zipper in yellow) stabilized sequence; Hairpin—relocation of the loop; and Motif loop—a shorter LP4 sequence that includes only the tip of the loop. The Hairpin and Motif loop were stabilized by disulfide bonds between two cysteines, as indicated by the orange dashed line. A linear peptide without the tryptophan (Trp) zipper; a linear peptide; a linear peptide eight amino acids shorter; and a cyclic peptide produced by the addition of cysteine to the C- and N-terminus of the peptide. (**C**) Illustration of CPP: Antp. Following the fusion of the Antp, the activity of various peptides in the induction of cancer cell death is indicated below each peptide (+ active; − non-active). (**D**) Tf-D-LP4, a loop-shaped VDAC1-based peptide stabilized by a Trp zipper and fused to the CPP Tf (pink); and Retro-Tf-D-LP4, a VDAC1-based peptide and Tf as CPP, all in retro-inverso order. In these peptides, the amino acids of the VDAC1-derived sequence are in a D configuration. These two peptides were highly active in cell death induction [5,7].

**Figure 4 biomolecules-14-01139-f004:**
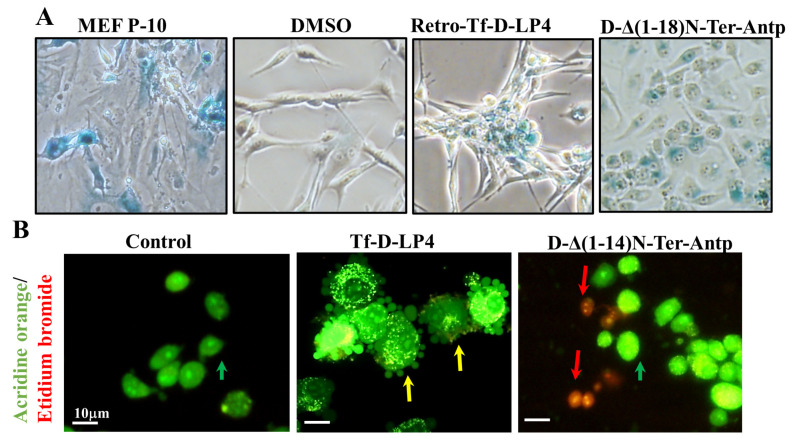
VDAC1-based peptides induce cellular senescence and apoptosis. (**A**) U-87MG cells (human glioblastoma) treated with Retro-Tf-D-LP4 peptide or with D-Δ(1–18)N-Ter-Antp (7 μM,12 h), and then subjected to β-gal activity (blue). Mouse embryonic fibroblast cells at passage 10 (MEF P-10) served as a positive control of senescent cells. The images were captured at 40×. (**B**) U-87MG cells treated with Tf-D-LP4 or with D-Δ(1–14)N-Ter-Antp (5 μM,3 h), and then stained with acridine orange and ethidium bromide (100 μg/mL). Green, yellow, and red arrows indicate live cells, cells with membrane blebbing (early apoptotic state), and late apoptotic states, respectively.

**Figure 5 biomolecules-14-01139-f005:**
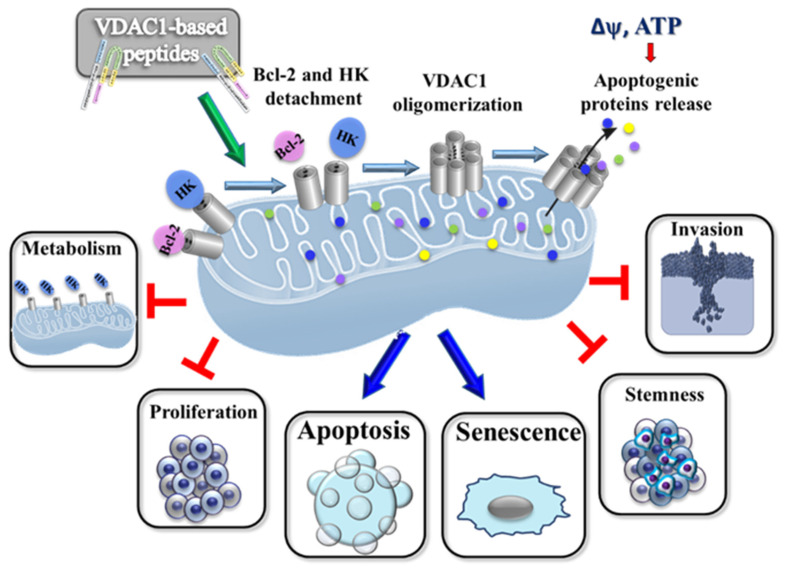
Mode of action of VDAC1-based peptides. In the mitochondria of cancer cells, VDAC1 is overexpressed and associated with proteins such as HK and Bcl-2. The cell is in homeostasis with respect to maintaining membrane potential (Δψ) and energy production and is protected from apoptosis. Following VDAC1-based peptide treatment, VDAC1-associated proteins such as HK and Bcl-2 interact with the peptides and detach from VDAC1. The disassociation of the proteins from VDAC1 leads to Δψ dissipation, decreased ATP production, mitochondrial dysfunction, VDAC1 oligomerization, and the release of Cyto c and other pro-apoptotic proteins—events that ultimately lead to cell death and/or senescence. Tumor treatment with VDAC1-based peptides results in attacks on hallmarks of cancer and reversal of oncogenic properties. These include mitochondrial dysfunction, decreased energy and metabolite production, arrested cell proliferation, induction of apoptosis, and inhibition of invasion and stemness.

**Figure 6 biomolecules-14-01139-f006:**
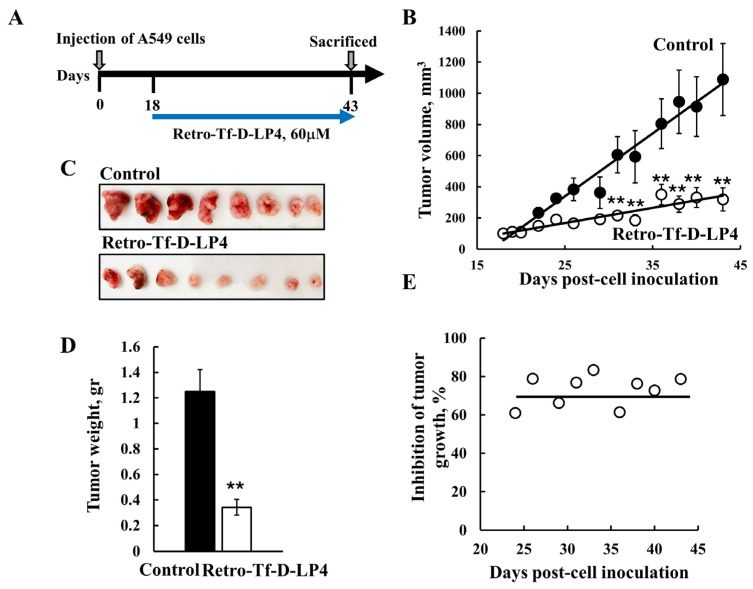
VDAC1-based peptide R-Tf-D-LP4 inhibits tumor growth in a lung cancer xenograft mice model. (**A**) Graphical representation of the xenograft experiment protocol used. A549 cells were inoculated subcutaneously (s.c.) into 8-week-old female nude mice (3 × 10^6^ cells/mouse). The tumor sizes were measured and the volume was calculated. (**B**–**D**) The R-Tf-D-LP4 peptide inhibits tumor growth. (**B**) On day 19, when tumor volume was 100–110 mm^3^, mice were sub-divided into two matched groups (eight mice/group), and injected every two days with DMSO (●, control, 0.26%) or R-Tf-D-LP4 (○, 60 μM). The calculated average tumor volumes are presented as means ± SE (*n* = 8), (** *p* ≤ 0.001). After 43 days, the tumors were dissected, photographed (**C**), and weighed (**D**). (**E**) Quantification of inhibition of tumor growth, showing constant inhibition over time.

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
