# Peer review of "VDAC1-Based Peptides as Potential Modulators of VDAC1 Interactions with Its Partners and as a Therapeutic for Cancer, NASH, and Diabetes"

_biomolecules, 2024, doi:10.3390/biom14091139_

Round 1

Reviewer 1 Report

Comments and Suggestions for Authors

This review presents current knowledge related to the voltage-dependent anion channel (VDAC1) as a multi-functional mitochondrial protein that acts in regulating both cell life and death. VDAC1 interacts with over 150 proteins and thus, can mediate and regulate the integration of mitochondrial functions with cellular activities. To target these protein–protein interactions, VDAC1-derived peptides have been developed mainly by the authors of this review. The authors demonstrated that these peptides are acting very rapidly and at low micromolar concentrations. VDAC1-based peptides indeed interfere with VDAC1 interacting proteins causing energy and metabolism impairment, interference with the action of anti-apoptotic proteins, and triggering of senescence and cell death. The peptides' activity has been tested both in cell cultures and validated in several mouse cancer models of glioblastoma, lung, and breast cancers. These peptides were also found to be effective in treating non-alcoholic fatty liver disease and diabetes mellitus. Thus, the authors suggest that VDAC1-based peptides targeting its’ interacting proteins, offer a new therapeutic approach for treating human diseases.

This is a detailed, comprehensive, and important review; However, there are major concerns that should be addressed as the followings:

- The general structure/outline of the review: The manuscript starts with the first chapter being “Protein-protein interactions as a target (a target for what???....the sentence is incomplete…); however, this is not the ‘natural’ place of this information; should appear at the end of the review; suggested in a new “conclusion” chapter; demonstrating similar reviews/studies for inhibiting protein-protein interactions. It should provide an update of the literature dealing not only with peptides; but also, with other existing ways for inhibiting those interactions, such as small molecules, antibodies and more.

The authors should also describe limitations and challenges still existing in developing peptides for inhibiting protein-protein interactions (and there are still quite a substantial number of them…).

In this regard lines 69-88 are completely irrelevant, as the reference cited (no. 7) for all the data provided, is from 2013 (!!!), over a decay before….

The review should start with a short paragraph describing indeed protein-protein interaction as a target for intervention and jumping directly to VDAC1 (chapter 2).

Chapter 3 cannot stand as a separate chapter and should be combined with chapter 2.

Please refer to Figure 1 in the new combined chapter (2&3; chapter 1 in the suggested revised review); its missing in the current version of the manuscript.

Chapters 5-7: there are numerous repeats in these 3 chapters about VDAC1 interactions with proteins (for example HK, Bcl-2, Bcl-xL and many more….); making this manuscript too long and hard to follow. Therefore, the authors should combine Chapters 5& 6 into one chapter; chapter 5 (proposed mode of action of VDAC-based peptides) being the lead and incorporating the additional information that appears in the original chapter 6. Accordingly, the review must be shortened substantially.

- Figure 1: This figure represents VDAC1 as a hub protein with its associated proteins. The authors should describe in the legend to the Fig. how was this map of interactions, the VDAC1 Interacome, created. Moreover, the authors should add a table (as a supplementary material) detailing each protein interaction and provide a reference(s) to support such data.

- Figure 2: It is highly recommended to add the tertiary structure of VDAC1 for better understanding the relative localization of the positions the peptides bind to.

- Self citation: Although the authors are major contributors in the research field of VDAC1; this review is citing mainly their own publications; and it should be reduced substantially.

Technical corrections:

- Lines 216-223 and lines 247-254 are repeats; please delete one of them.

- Chapter 7.3: Lines 602-608 are in italic writing; please change it to normal word writing.

Author Response

Reviewer 1

 Comments and Suggestions for Authors

We thank this reviewer for pointing out that: " This is a detailed, comprehensive, and important review; ". We sincerely appreciate the insight and comments that have been used to revise our manuscript in accordance with the referee's specific comments. We have incorporated the suggested changes to the revised manuscript as detailed below and are marked in blue color in the revised version.

  1. - The general structure/outline of the review:The manuscript starts with the first chapter being “Protein-protein interactions as a target (a target for what???....the sentence is incomplete…); however, this is not the ‘natural’ place of this information; should appear at the end of the review; suggested in a new “conclusion” chapter; demonstrating similar reviews/studies for inhibiting protein-protein interactions. It should provide an update of the literature dealing not only with peptides; but also, with other existing ways for inhibiting those interactions, such as small molecules, antibodies and more.

1A. This is a good suggestion; we have removed the first chapter on protein-protein interactions (PPIs) and instead we  included an overview where PPIs were presented in 2-3 sentences.

  1. The authors should also describe limitations and challenges still existing in developing peptides for inhibiting protein-protein interactions (and there are still quite a substantial number of them…).

As suggested, we added a Conclusion chapter at the end of the MS, which shortly presents the PPIs, targeting VDAC1 PPIs, and the peptides advantages, challenges and about therapeutic peptides: current applications and future directions as well as the recent advances in the development of protein–protein interactions modulators: mechanisms and clinical trials.

In this regard lines 69-88 are completely irrelevant, as the reference cited (no. 7) for all the data provided, is from 2013 (!!!), over a decay before….

 As the whole section was removed, these lines were also removed and updated references for the peptides as a drug were added in the Conclusion chapter.

The review should start with a short paragraph describing indeed protein-protein interaction as a target for intervention and jumping directly to VDAC1 (chapter 2).

We have addressed this suggestion as indicated above.

  1. Chapter 3 cannot stand as a separate chapter and should be combined with chapter 2.

Please refer to Figure 1 in the new combined chapter (2&3; chapter 1 in the suggested revised review); its missing in the current version of the manuscript.

The suggested, chapter 2 and 3 were combined and Fig. 1 is now indicated.

  1. Chapters 5-7: there are numerous repeats in these 3 chapters about VDAC1 interactions with proteins (for example HK, Bcl-2, Bcl-xL and many more….); making this manuscript too long and hard to follow. Therefore, the authors should combine Chapters 5& 6 into one chapter; chapter 5 (proposed mode of action of VDAC-based peptides) being the lead and incorporating the additional information that appears in the original chapter 6. Accordingly, the review must be shortened substantially.

We found that it is better to combine chapter 5 with chapter 2 which deals with VDAC1 interactome as summarized in Fig. 1. We shortened the text by over 30% and removed several citations.

  1. - Figure 1:This figure represents VDAC1 as a hub protein with its associated proteins. The authors should describe in the legend to the Fig. how was this map of interactions, the VDAC1 Interactome, created. Moreover, the authors should add a table (as a supplementary material) detailing each protein interaction and provide a reference(s) to support such data.

 We published about VDAC1 and its interacting proteins in detail in two reviews (Refs. 2, and 3).  Ref. 2 includes a table with the method used and the citation for it.  As suggested, we added to the figure 1 legend the major methods used to demonstrate these interactions.

- Figure 2: It is highly recommended to add the tertiary structure of VDAC1 for better understanding the relative localization of the positions the peptides bind to.

 As suggested, we added the VDAC1 3D-structure and labeled in it the 2 sequences identified to be involved in protein-protein interaction, to this we added part of Fig. 2, and for simplicity we rearranged the different versions of the peptides developed in a separated figure (now Fig. 3).

- Self citation: Although the authors are major contributors in the research field of VDAC1; this review is citing mainly their own publications; and it should be reduced substantially.

 Now we have 147 citations instead  of 188 in submitted version. Thus, we have removed about 41 references including 13 self-citations.

It is important to note that most of the studies related to VDAC1-based peptide, which is central to this MS, were conducted in my lab over the past 20 years,

Technical corrections:

- Lines 216-223 and lines 247-254 are repeats; please delete one of them.

 Thank you, the repeated part has been removed

- Chapter 7.3: Lines 602-608 are in italic writing; please change it to normal word writing.

Done

Reviewer 2 Report

Comments and Suggestions for Authors

Author Response

Reviewer 2

We thank this reviewer for correctly summarizing the aims of this review.  We sincerely appreciate the insight and comments that have been used to revise our manuscript in accordance with the referee's specific comments. We have incorporated the suggested changes to the revised manuscript as detailed below and are marked in blue color in the revised version.

  1. On page 3, line 99, Ca2+, please check the charge and correct it according to publication requirements, e.g., superscript.

Done

  1. Page 3, line 100, ‘It mediates mitochondria–endoplasmic reticulum (ER) cross-talk and is involved in inflammasome activation and more.’ This sentence wasn't clear; what is meant by more? Make it clear.

This sentence is now removed

  1. Page 3, line 129, remove the paragraph between lines 128 and 129.

Done

  1. Page 3, line 140, correct ‘givin that’ to ‘Given that’

Done

  1. Check the notation for amino acids D and L; it should be in a smaller font. Make the corrections according to the manuscript

In most Publications the amino acids conformations L and D ate in capital letters

. 6. Page 5, line 212. ‘Furthermore, the peptide was found to be effective in treating nonalcoholic fatty liver disease and type 2 diabetes.’ Correct the sentence: which peptide? It was not clear here to readers

Done

  1. Page 8, line 300, remove the comma after ‘The’.

Done

  1. Figures 2 and 4 are not clear, make it clear. The figure 2 looks congested; please simplify it.

We agree with this reviewer, we have made Fig. 2 as 2 figures, added the VDAC1 3D-structure with the 2 sequences identified to be involved in protein-protein interaction labeled in it.  To this we added part of Fig. 2 and for simplicity we rearranged the different versions of the peptides developed in a separated figure (now Fig. 3)

  1. Cited references are fine

Reviewer 3 Report

Comments and Suggestions for Authors

Reviewer response

In this manuscript (biomolecules-3149820), the authors reviewed the voltage-dependent anion channel 1 (VDAC1) and its role and the therapeutic based peptides against cancer, NASH, and diabetes.  However, the current version requires extensive checking, and some stylistic and grammatical issues as pointed out in the specific comments below. The authors must improve the review by organizing the contents.

Concerns

1.     The address of the authors is not written completely.

2.     “Aberrant PPIs are associated with various diseases, including cancer, as well as infectious and neurodegenerative diseases.” Please add references.

3.     “VDAC1 interacts with over 150 proteins that regulate the integration of mitochondrial functions with other cellular activities [8-10,21]”. Repeated in Section 3. It is better to write these two sections together.

4.     In section 4, “Givin that VDAC1 can interact with so many proteins, we developed VDAC1-based peptides [22-25,33,34] as a “decoy” to compete with “. Please correct it.

5.     Section 4.2 is repeated. Please do it correctly.

6.     “Treatment of tumors with VDAC1-based peptides effectively eliminated these cells in glioblastoma, lung, and breast cancers, as evidenced by significant reductions in the expression levels of cancer type-specific stem cell markers [24,33,34]” Please mention VDAC based peptides.

7.     “Bcl-2 and Bcl-xL were shown to interact with bilayer-reconstituted VDAC1 and to, subsequently, reduce the channel conductance of native, but not mutated, VDAC1, as well as to protect against apoptosis in cells expressing native, but not mutated, VDAC1 [22,25,67,121,127-129].” Rewrite the sentence to make it more understandable.

8.     “It was also hypothesized [104] that the interaction of TSPO with VDAC1 contributes to the efficiency of mitochondrial quality control, regulating mitochondrial structure and function [105,106]”. Describe ref 104.

9.     In the three types glioblastoma, lung, and breast cancers, the peptides induced massive apoptotic cell death and induced the overexpression of key proteins in apoptosis, including caspases 3, 8, p53, Cyto c, and SMAC/Diablo, while decreasing the expression of antiapoptotic proteins, such as Bcl-2 [26,39].” Write 8 as caspase 8.

10.  Conclusion of the review can be included.

Comments on the Quality of English Language

English language can be improved. 

Author Response

We sincerely appreciate the insight and comments that have been used to revise our manuscript in accordance with the referee's specific comments. We have incorporated the suggested changes to the revised manuscript as detailed below and are marked in blue color in the revised version.

Concerns

  1. The address of the authors is not written completely.

 Corrected

  1. “Aberrant PPIs are associated with various diseases, including cancer, as well as infectious and neurodegenerative diseases.” Please add references.

This sentence is now part of the Conclusions chapter, and references were added.

  1. “VDAC1 interacts with over 150 proteins that regulate the integration of mitochondrial functions with other cellular activities [8-10,21]”. Repeated in Section 3. It is better to write these two sections together.

 We agree with this comment, we rearranged the  sections related to the same topic, we combined chapter 5 with chapter 2 which deals with VDAC1 interactome as summarized in Fig. 1.

4. In section 4, “Givin that VDAC1 can interact with so many proteins, we developed VDAC1-based peptides [22-25,33,34] as a “decoy” to compete with “. Please correct it.

Corrected

  1. Section 4.2 is repeated. Please do it correctly.

Thanks, it is removed

  1. “Treatment of tumors with VDAC1-based peptides effectively eliminated these cells in glioblastoma, lung, and breast cancers, as evidenced by significant reductions in the expression levels of cancer type-specific stem cell markers [24,33,34]” Please mention VDAC based peptides.

The used peptide is now indicated

  1. “Bcl-2 and Bcl-xL were shown to interact with bilayer-reconstituted VDAC1 and to, subsequently, reduce the channel conductance of native, but not mutated, VDAC1, as well as to protect against apoptosis in cells expressing native, but not mutated, VDAC1 [22,25,67,121,127-129].” Rewrite the sentence to make it more understandable.

It has been now changed to: Bcl-2 and Bcl-xL were found to interact with bilayer-reconstituted VDAC1, leading to a decrease in the channel conductance of native VDAC1, but not its mutated form [21]. Additionally, these proteins provided protection against apoptosis in cells expressing native VDAC1, but not in those with the mutated version [19,21,28,59,60] (section 2.2.)

  1. “It was also hypothesized [104]that the interaction of TSPO with VDAC1 contributes to the efficiency of mitochondrial quality control, regulating mitochondrial structure and function [105,106]”. Describe ref 104.

This section was modified and moved to section 2, together with other VDAC1 interacting proteins (section 2.4, page 4.) Also included in a new section:

Section 5.4 VDAC1-based peptide targeting the levels of testosterone

  1. In the three types glioblastoma, lung, and breast cancers, the peptides induced massive apoptotic cell death and induced the overexpression of key proteins in apoptosis, including caspases3, 8, p53, Cyto c, and SMAC/Diablo, while decreasing the expression of antiapoptotic proteins, such as Bcl-2 [26,39].” Write 8 as caspase 8.

Done

  1. Conclusion of the review can be included.

 We added a Conclusion chapter at the end of the MS, which shortly presents the PPIs, targeting VDAC1 PPIs, and the peptides advantages, challenges and about therapeutic peptides: current applications and future directions as well as the recent advances in the development of protein–protein interactions modulators: mechanisms and clinical trials.

Round 2

Reviewer 1 Report

Comments and Suggestions for Authors

The authors addressed most of the suggestions and corrected the manuscript accordingly. 

Author Response

  Comment: The authors addressed most of the suggestions and corrected the manuscript accordingly.

We thank this reviewer 

Reviewer 3 Report

Comments and Suggestions for Authors

Concerns

1.      In line 74, add space before “and”.

2.      Line 84, please correct the sentence.

3.      Line 146, please format the sentence by adding space before “next section 2.3.”

4.      In line 219, check the spelling of measuring.

5.      In line 375, check the spelling of influencing

6.      Line 556, please correct the sentence.

7.      Please follow the same format for figure (either figure or fig).

8.      Before publishing, please rewrite the sentences that are copied from the other research papers.  

Comments on the Quality of English Language

English language must be improved. 

Author Response

Concerns

  1. In line 74, add space before “and”.

 Done

  1. Line 84, please correct the sentence.

Done

  1. Line 146, please format the sentence by adding space before “next section 2.3.”

Done

  1. In line 219, check the spelling of measuring.

Spelling is now corrected

  1. In line 375, check the spelling of influencing

Spelling is now corrected

  1. Line 556, please correct the sentence.

The sentence : PDX1 is crucial for insulin gene expression and islet development, function, and proliferation, suggesting potential benefits of the VDAC1-based peptide as a treatment for diabetes [103].

 It is now modified to:

PDX1 is essential for regulating insulin gene expression and is also key to the development, function, and proliferation of pancreatic islets. his underscores the potential benefits of using the VDAC1-based peptide as a treatment for diabetes.

  1. Please follow the same format for figure (either figure or fig).

Figure is used in the title of each figure and throughout the text is referenced in a bracket, as Fig.  with the number.

  1. Before publishing, please rewrite the sentences that are copied from the other research papers.  

 Apologies if it seemed like I copied any sentences from other publications. Considering that this review is on a topic I developed and published over 16 years, it’s possible that my way of presenting the issue or topic is consistent across this and our previous publications.